# COVID-19 among People Living with HTLV-1 Infection in Rio de Janeiro, Brazil

**DOI:** 10.3390/pathogens12020242

**Published:** 2023-02-03

**Authors:** Marzia Puccioni-Sohler, Alana Cristina Jasset Miranda, Cíntia da Silva Mello, Stéphanie Monnerat Magalhães, Luciane Cardoso dos Santos Rodrigues, Dario J. H. P. Signorini

**Affiliations:** 1Escola de Medicina e Cirurgia, Universidade Federal do Estado do Rio de Janeiro, Rio de Janeiro 20271-062, RJ, Brazil; 2Post-Graduation in Infectious and Parasitic Diseases, Faculty of Medicine, Universidade Federal do Rio de Janeiro, Rio de Janeiro 21941-913, RJ, Brazil; 3Immunology Laboratory, Hospital Universitário Gaffrée e Guinle-Ebserh, Universidade Federal do Estado do Rio de Janeiro, Rio de Janeiro 20271-062, RJ, Brazil

**Keywords:** COVID-19, SARS-CoV-2, HTLV-1, HTLV-1-associated myelopathy, anxiety, post-COVID-19, vaccination

## Abstract

The impact of coronavirus disease 2019 (COVID-19) on people living with human T-cell leukemia virus type 1 (HTLV-1) is unknown. The aim of this study is to evaluate the COVID-19 risk factors and outcomes of HTLV-1-infected individuals. A retrospective study of seropositive HTLV-1 outpatients seen during the COVID-19 pandemic period (2020–2022) was conducted in a Tertiary Hospital in Rio de Janeiro, Brazil. We compared the demographic and comorbidity/risk factors in patients with COVID-19 and non-COVID-19 diagnoses. In addition, the clinical features of COVID-19 and vaccination status were also investigated in 51 HTLV-1-infected individuals. The majority (88.2%) had COVID-19 comorbidity/risk factors. Seven cases were vaccinated against COVID-19. Overall, 19 out of 51 (37.3%) individuals were diagnosed with COVID-19. We found differences only in the frequency of anxiety in both groups: 57.9% in the COVID-19 group vs. 15.6% in the non-COVID-19 (*p* < 0.05) group. Thirteen out of nineteen (68%) of the COVID-19 cases progressed to mild/moderate illness, one remained asymptomatic, and 26.3% progressed to severe illness. All of the individuals recovered at home, but the majority (57.9%) developed post-COVID-19 symptoms: anosmia and ageusia (31.6%), worsening anxiety (15.8%), and a feeling of pain in the legs (15.8%). The patients with post-COVID-19 conditions were unvaccinated. Our findings show that HTLV-1 did not increase the risk of lethal COVID-19 and underline the importance of promoting mental health in HTLV-1-infected individuals.

## 1. Introduction

The SARS-CoV-2 (severe-acute-respiratory-syndrome-related coronavirus 2) is a virus with a single-stranded RNA genome, belonging to the coronaviridae family, first isolated in the city of Wuhan (China) in December 2019 [1,2,3]. In 2020, the viral infection and the resulting disease, coronavirus disease 2019 (COVID-19), progressed to a pandemic [1,4]. Transmission between humans occurs by respiratory droplets, by direct contact between infected individuals, objects, and surfaces [1,2,3]. The incubation period is between 1 and 14 days [1]. COVID-19 is a multisystem infectious disease that mainly affects the respiratory tract, with a high potential for transmissibility and lethality and global distribution [1]. Since 2020, more than 2600 SARS-CoV-2 lineages have been identified worldwide through genome sequencing [5]. In Brazil, from March 2020 to January 2022, more than 200,000 genomes were sequenced, which corresponds to an index of 569.5 for every 100,000 confirmed cases of COVID-19 infection [6]. The variants of concern, that is, the variants most found in the sequenced samples, have been changing since the beginning of the pandemic. In 2020, the B.1.133 variant was the most identified in Brazil; in 2021, P.2, Gamma (P.1) and Delta (B.1.617.2) variants were more prevalent; whereas in 2022, different strains of the Omicron variant (BA.1, BA.2, BA.4, BA.5, BE.9, BQ.1, BQ.1.1, DL.1, XBB) were the most commonly found in the sequencing [6,7]. These results are similar to those found in our research location, the state of Rio de Janeiro [6]. On the other hand, from January 2020 to 16 December 2022, more than 656 million cases of COVID-19 were reported worldwide, with 6,668,259 deaths [4]. As of 15 December 2022, 35,809,832 cases of COVID-19 and 691,652 deaths were reported in Brazil [8], which was one of the countries with the most cases of COVID-19 in the world. In the city of Rio de Janeiro, as of 16 December 2022, more than 1,271,267 cases were confirmed, with 37,908 deaths [9]. Although most individuals with COVID-19 are asymptomatic (33%) or with mild disease (81%), 20% need hospitalization, given that 5–8% evolve with severe forms and need to be admitted to an intensive care unit (ICU) [10,11]. By the end of 2020, several vaccines were available for use in different parts of the world [12]. The vaccination in Brazil started on January 2021.

The human T-cell lymphotropic virus (HTLV-1) was the first retrovirus described in humans, in 1980 [13]. HTLV-1 has become endemic in several regions: South America, Africa, the Caribbean, and Japan, affecting more than 5 to 10 million individuals worldwide [14]. It is estimated that there are approximately 800,000 to 2.5 million HTLV-1-infected individuals in Brazil [14]. The virus can be transmitted through unprotected sexual intercourse, contact with infected blood, organ transplantation, during childbirth, transplacentally, and through breastfeeding [14]. Most infected individuals are asymptomatic and 5% may develop adult T-cell leukemia/lymphoma (ATLL) and/or chronic inflammatory diseases such as HTLV-1-associated myelopathy (HAM) [13,14,15]. HAM is a disabling progressive demyelinating myelopathy whose main feature is spastic paraparesis [15]. Other main inflammatory disorders associated with HTLV-1 infection include arthritis, polymyositis, uveitis, infective dermatitis, and Sjogren’s syndrome [14].

Individuals infected with HTLV-1 tend to have a higher occurrence of co-infections with other infectious agents (helminthic, viral, and bacterial) in comparison with the general population [16]. This occurs due to the changes in the immune response induced by the persistent HTLV-1 infection. Likewise, SARS-CoV-2 causes an exacerbated generalized inflammatory response, which, added to risk factors and comorbidities, tends to evolve into a poor prognosis [17,18]. However, the impact of COVID-19 in people living with HTLV-1 is still unknown. It can be hypothesized that individuals infected with HTLV-1 would have a higher risk of developing lethal COVID-19 [19,20]. The present study assesses the risk factors associated with COVID-19 and its clinical outcomes among individuals living with HTLV-1 in a Brazilian endemic area, Rio de Janeiro city.

## 2. Material and Methods

### 2.1. Study Design

This is a retrospective study from a cohort of HTLV-1 patients seen in a tertiary hospital (Hospital Universitário Gaffrée e Guinle-Ebserh/Universidade Federal do Estado do Rio de Janeiro), Rio de Janeiro, Brazil. All patients were HTLV-1-positive at the time of COVID-19 exposure.

### 2.2. Data Source

Patient data were collected using a structured questionnaire for HTLV-1 adults (≥18 years old), seen in the Neuroinfection Outpatient clinic from 1 January 2020 to 19 October 2022. The questionnaire included demographic and clinical data, comorbidity/risk factors, vaccination status, and outcomes for COVID-19. All subjects underwent reactive HTLV-1 serological screening by enzyme-linked immunosorbent assay (Murex HTLV-I + II, Diasorin, UK) confirmed by Western blot (HTLV BLOT 2.4—Genelabs Diagnostics^®^, Science Park, Singapore). The HTLV-1-infected individuals were composed of two groups: asymptomatic HTLV-1 carriers and patients with HAM according to Osame’s criteria (1990) [15].

### 2.3. COVID-19 Variables Definition

COVID-19 cases were categorized as confirmed or asymptomatic. COVID-19 was confirmed in cases of influenza-like illness or severe acute respiratory syndromes by: (i). clinical criteria, when associated with acute anosmia or ageusia; (ii) clinical–epidemiological criteria, with a history of close or home contact in the 14 days prior to the appearance of signs and symptoms with a confirmed case for COVID-19; (iii) clinical–imaging criteria, with the presence of ground-glass opacity on tomography of the lungs; and (iv) laboratory criteria, with positive viral nucleic acid test (real-time PCR) or antigen (immunochromatographic) results for SARS-CoV-2 on throat swab samples. In symptomatic non-vaccinated COVID-19, the laboratory criteria also include reactive serology for IgM, IgA, and/or IgG (enzyme-linked immunosorbent assay, chemiluminescence immunoassay, immunochromatographic) [21]. Asymptomatic was defined by the absence of signs and symptoms with a positive laboratory test for SARS-CoV-2 (positive viral nucleic acid test or antigen) [21].

The severity of COVID-19 symptoms was defined as mild, moderate, severe, or critical. Mild severity is characterized by the presence of nonspecific signs and symptoms such as cough, pain throat, or runny nose, followed or not by anosmia and ageusia. The moderate form was determined by the progression of a mild clinical picture of COVID-19 to adynamia, diarrhea, hyporexia, prostration, and the presence of pneumonia without signs or symptoms of severity (absence of respiratory failure). The severe case included cyanosis, dyspnea, and oxygen saturation below 95%, progressing to severe acute respiratory syndrome. The critical form of the disease consisted of multiple organ dysfunction (acute heart, liver and kidney failure, cardiogenic shock, myocarditis, or cerebrovascular accident), severe respiratory failure, admission to ICU, the need for respiratory support, severe pneumonia, sepsis, and acute respiratory distress syndrome [21].

The comorbid/risk factor conditions included previous neurological (HAM) or cardiovascular disorder/hypertension (systolic artery pressure > 140 mmHg), sedentary lifestyle, diabetes, smoker (≥one cigarette/day for ≥1 year), anxiety, and depression, as reported elsewhere [22,23,24,25]. In addition, prevention measures (e.g., the use of masks and alcohol gel) and the vaccination status were also investigated.

Post-COVID-19 conditions were characterized by the continuation or emergence of new symptoms three months after a confirmed SARS-CoV-2 infection [26].

### 2.4. Ethics Statement

This study was approved by the Hospital Universitário Gaffrée e Guinle-Ebserh (HUGG-Ebserh/UNIRIO) Ethical Committee Board, Rio de Janeiro, Brazil. Individual informed consent was not required for this study. The researchers maintained strict confidentiality regarding the identity of participants and data obtained.

### 2.5. Statistical Analysis

Medical record information was included in a Microsoft Excel spreadsheet. Descriptive statistics analysis was based on means, standard deviation, medians, proportions, ranges, and percentages (%). The inferential analysis consisted of Student’s *t*-test for independent samples and the chi-squared (χ2) or Fisher’s exact test. Normality in data distribution was assessed using the Shapiro–Wilk test. The significance was determined by a set at 5%. The statistical analysis was processed using SPSS version 26 (SPSS, Chicago, IL, USA).

## 3. Results

### 3.1. Characteristics of HTLV-1-Infected Individuals

Data from 51 out of 52 HTLV-1 seropositive cases were included in the analysis. One HAM case of familial exposition was excluded since she did not meet the criterion used for COVID-19 [21]. The average (range) age of 51 HTLV-1 cases was 54.7 ± 13.9 (26–79) years old. There were 36 females (70.6%) and 15 males (29.4%). The race/ethnicity distribution included 29 cases of African descendent, 21 White, and one of Asian descendent. The majority (63%) had neurological chronic disease associated with HTLV-1 infection (HAM). The other 19 (37%) were asymptomatic HTLV-1 carriers. Comorbidity/risk factors were reported in 88.2% (45/51) of the participants. The main conditions consisted in sedentary lifestyle (64.7%), anxiety (31.4%), cardiovascular disease (19.6%), obesity (17.6%), depression (11.8%), diabetes mellitus (11.8%), and chronic obstructive lung disease (7.8%). All reported the use of prevention measures (mask and social distancing). A total of nine HTLV-1-infected individuals were seen in 2020 and the remaining 42 were seen during the vaccination period (2021–2022). A total of 7 (17%) out of these 42 were COVID-19-vaccinated. From the vaccinated individuals, there was one man and six women. In relation to 32 HAM patients, 18 (56%) were in long-term use of corticosteroids (prednisone 5 mg daily) during the study period [27].

### 3.2. Main Risk Factors Associated with COVID-19 in HTLV-1 Infection

Out of 51, 19 (37.3%) had COVID-19: 18 confirmed symptomatic and 1 asymptomatic. The characteristics of COVID-19 and non-COVID-19 cases are shown in Table 1. Although we found a higher frequency of smoking (40 vs. 8.3%) and age ≥ 60 years (66.7 vs. 30.6%) and lower obesity (0 vs. 25%) in males than in females, respectively (*p* < 0.05), there was no difference in the COVID-19 frequency in the gender distribution (*p* > 0.05). There was no significant association between COVID-19 and other variables such as demographic (age, race); previous neurological (HAM), lung, or cardiovascular disorders; diabetes mellitus; depression; obesity; or sedentary lifestyle (*p* value > 0.05), but there was for the presence of anxiety (*p* < 0.05). There was no statistical difference in the frequency of COVID-19 between HAM patients treated with corticosteroids (n = 6) in comparison to the non-treated (n = 4) group (*p* > 0.05).

Three (15.7%) vaccinated patients had COVID-19, in comparison to four (12.5%) vaccinated in the non-COVID-19 group (*p* > 0.05). One of the COVID-19 vaccinated cases remained asymptomatic (Table 1).

### 3.3. Diagnosis and Clinical Outcome in Confirmed COVID-19

The COVID-19 diagnosis was based only on clinical criteria in 15.7% (3/19), clinical–epidemiological criteria in 15.7% (3/19), clinical–laboratory criteria in 21% (4/19), clinical–epidemiological–imaging criteria in 5% (1/19), clinical–epidemiological–laboratory criteria in 31% (6/19), clinical–epidemiological–laboratory–imaging criteria in 5% (1/19), and only laboratory criteria in one (5%) asymptomatic case. Twelve (63.2%) patients had laboratory confirmation: nine had a positive PCR for SARS-CoV-2 on throat swab and three were symptomatic non-vaccinated COVID-19 cases with reactive serology.

Among the nineteen COVID-19 cases, one was asymptomatic, thirteen (68.4%) had mild/moderate COVID-19, and five (26.3%) presented severe COVID-19 due to the presence of dyspnea and an oxygen saturation below 95%. The main manifestations in the 18 symptomatic COVID-19 patients were respiratory symptoms (dyspnea), ageusia, myalgia, fatigue, and anosmia. All were treated at home. There were no patients with critical cases or hospitalization in intensive care units (Table 2).

All patients recovered after COVID-19, but the majority 11/19 (57.9%) developed post-COVID-19 conditions such as: the presence of anosmia, ageusia, and worsening of anxiety. Pain, burning, or a subjective sensation of heaviness in the lower limbs, not compatible with restless legs syndrome, were also reported in two patients with HAM and one HTLV-1 carrier (Table 3). From the total COVID-19 group (n = 19), 77% (7/9) of the cases had post-COVID-19 conditions in 2020 and 40% (4/10) in 2021–2022. The patients presented with 1–6 (median, 1) symptoms. Six out of eleven (54%) cases of post-COVID-19 had HAM, and the other five were HTLV-1 asymptomatic carriers. None of the 11 patients with post-COVID-19 conditions were vaccinated in comparison to 37.5% (3/8) vaccinated individuals from the group without post-COVID-19 conditions (*p* < 0.05).

## 4. Discussion

Here, we report a retrospective observational study of 51 Brazilian adults infected with HTLV-1 and identified the main risk factors for COVID-19. Co-infections such as human immunodeficiency virus, hepatitis B and C virus, syphilis, tuberculosis, strongyloidiasis, and scabies are frequent in HTLV-1-infected individuals [16]. A total of 37.3% of the HTLV-1-infected individuals in our study were found to have SARS-CoV-2 co-infection. COVID-19 comorbidity/risk factors were very frequent (88.2%) in the studied group. The main risk factor associated with COVID-19 was anxiety. Despite the high frequency of comorbidities/risk factors, the illness was not severe in the majority (73.6%) of the cases. The patients with severe COVID-19 were not hospitalized or in need of intubation, and none died. Post-COVID-19 conditions were observed in the majority (57.9%) of the cases, predominantly anosmia and ageusia, worsening anxiety, and pain in the lower limbs.

In 2020, after the rapid spread of SARS-CoV-2, it was observed that there were specific groups at greater risk of developing the infection and evolving with a worse prognosis for COVID-19 [1,3,18]. Among these, there were individuals aged over 60 years and with comorbidities, such as: arterial hypertension, cancer, chronic kidney disease, chronic liver disease, chronic lung diseases, dementia or other neurological conditions, diabetes mellitus, Down syndrome, cardiovascular disease, HIV, an immunocompromised status, psychiatric conditions, overweight and obesity, pregnancy, sickle-cell anemia or thalassemia, smoking, transplant recipients, stroke, drug abuse, and tuberculosis. Other risk factors are also thought to have an influence on the involvement of COVID-19 [28,29]. Males have a more fragile immune system, a lifestyle more associated with risk behaviors, and a higher number of angiotensin-converting enzyme 2 (ACE2) receptors in the pulmonary endothelium. Ethnic/racial disparities could likely be linked to socioeconomic causes associated with poorer hygiene conditions. In Brazil, higher COVID-19 severity (mortality) is also related to regions with lower socioeconomic development and, consequently, worse access to health systems [30]. On the other hand, HTLV-1 causes persistent infection in the body, predominantly infecting CD4+ T lymphocytes, where it remains in the DNA as a provirus. Individuals infected with HTLV-1 are at risk of opportunistic infections due to alterations in the immune functions of the host [16]. The main characteristics of our HTLV-1-seropositive individuals were a mean age of 51.7 ± 13.0, predominance of females, African descent, the presence of chronic neurological disease (HAM), and sedentary lifestyle. We did not find any difference between most comorbidity/risk factors (including gender and race distribution and the presence of chronic neurological disease associated with HTLV-1) in the groups with and without COVID-19. A previous study evaluated the profile of 51,383 cases of COVID-19 in the state of Rio de Janeiro in 2020. It showed that the predominant factors associated with death, which occurred in 40.5% of cases, were male gender, advanced age (>75 years), oxygen saturation <95%, respiratory distress, dyspnea, chronic kidney and neurological diseases, immunosuppression, and the use of ventilatory support [31]. Although the majority (64%) of our HTLV-1-infected individuals had HAM, which manifests predominantly in females, usually after the age of 40 [15], its distribution was similar in both groups (COVID-19), as well the majority of other comorbidities. We had no lethal case nor acute respiratory distress syndrome in any studied period (2020–2022). This may be explained by the fact that, considering that HTLV-1 causes chronic infection, our outpatients are followed up over time in the hospital. For this reason, most comorbidities are under constant medical control. The HUGG-Ebserh is a public university hospital located in Tijuca, a neighborhood in the north of Rio de Janeiro city, which has a quality-of-life index above the average for the municipality. However, the majority of our patients with HTLV-1 infection come from other municipalities (counties) with lower incomes.

Anxiety is a frequent manifestation among patients with HTLV-1 infection. Rocha-Filho et al. (2018) found that 63.08% of HTLV-1-infected patients presented anxiety in comparison to 24.4% in uninfected patients [32]. The anxiety that occurs in HTLV-1 infection is justified by the uncertainty of developing symptoms, which are sometimes disabling, such as HAM. In this condition, the patient can progress to paraplegia and become wheelchair-bound; therefore, it impairs the quality of life of the population [33]. On the other hand, HTLV-1 is still a neglected health problem [34]. In this context, the unified Brazilian health system does not carry out the HTLV-1 confirmatory test for asymptomatic carriers or HAM. It may cost up to USD 600 in private laboratories [34]. In addition, the COVID-19 pandemic had a worldwide psychological impact, a result of social isolation, loss of family and friends, and fear of illness and death. The consequence of this condition was an increase in mental illnesses, with a great influence on public health. A previous study estimated the emergence of more than 76 million cases of anxiety in the world due to the COVID-19 pandemic [35]. In addition to the pandemic environment, COVID-19 itself can cause changes in the mental health of the population [36]. Coronaviruses have a neurotropic potential that can induce changes in the central nervous system (CNS) through direct viral infection or the immunological response, targeting neurological interaction pathways [37]. SARS-CoV-2 induces an immune response with the local and systemic production of cytokines, chemokines, and other inflammatory markers [37]. In fact, we found a higher frequency of anxiety in the group of HTLV-1-infected patients with COVID-19.

In this sense, vaccination is extremely important as it reduces the risk of COVID-19. According to the World Health Organization (2022), vaccination against COVID-19 can save lives by providing strong protection against illness, hospitalization, death, and post-COVID-19 conditions [38]. In our study, only 17% were COVID-19 vaccinated, the majority being females (6/7). Our patients justified their initial resistance to vaccination due to the fear of vaccine complications associated with the high circulation of false information on the subject. All recovered, but the majority (57.9%) developed post-COVID-19 syndrome. All of our patients with post-COVID-19 conditions were unvaccinated.

Post-COVID-19 syndrome occurs after asymptomatic SARS-CoV-2 infection or after critical illness, and it is independent of the initial severity of the infection [26]. According to the WHO (2022), 10 to 20% of individuals who are infected with SARS-CoV-2 may have post-COVID-19 syndrome [26]. Symptoms may fluctuate and relapse over time. Symptoms include: anxiety, depression, dyspnea, difficulty breathing, tiredness, fatigue, malaise after physical exertion, “brain fog” (difficulty concentrating and thinking; cognitive dysfunction), cough, chest pain, abdominal pain, cardiovascular changes (palpitations), joint or muscle pain, sleep disturbances, sensations of pins and needles, diarrhea, vertigo, skin irritation, mood swings, menstrual cycle changes, anosmia, and ageusia. These last for at least two months and are not explained by other diagnoses [26,39,40]. In addition, systemic symptoms (including in the heart, lungs, kidneys, skin, and brain) or autoimmune conditions may arise [26]. A study from China analyzed 1733 cases of individuals who contracted COVID-19 and observed that the majority (75%) had some complication after COVID-19, with fatigue/muscle weakness (63%), sleep disorders (26%), and anxiety (23%) developing [39]. A previous Brazilian study that monitored 646 in-home, outpatient, and hospitalized COVID-19-positive patients found that 50.2% presented with long-COVID syndrome. Most patients had two or three symptoms, with a predominance of fatigue (35.6%), persistent cough (34%), dyspnea (26.5%), mental disorders (20.7%), and anosmia and ageusia (20.1%). The presence of risk factors was determinant for the evolution of prolonged symptoms [41]. Another Brazilian analysis evaluated the outcome of 439 patients with COVID-19 discharged from the hospital between 1 July 2020 and 31 March 2021. The main symptoms of long-COVID-19 were fatigue (63.1%), dyspnea (53.7%), arthralgia (56.1%), and depression/anxiety (55.1%) [42]. Although our study involved a unique cohort of outpatients with HTLV-1 and COVID-19 co-infection, we also found a high frequency of post-COVID-19 conditions (57.9%) despite the good evolution of all cases (most with one symptom), without hospitalization or death. These complications occurred predominantly in 2020 (77% vs. 40%) in comparison to 2021–2022, respectively. In this initial pandemic period (2020), the greater severity of cases seemed to be more related to the absence of vaccine against a new virus. Similar to other Brazilian studies, the main post-COVID-19 conditions included anosmia and ageusia (31.6%) and worsening of anxiety (15.8%). Pain or burning or a subjective sensation of heaviness in the legs occurred in three patients. This may be a characteristic unique to HTLV-1-infected COVID-19 patients, not yet reported elsewhere. In addition, our patients reported headache, sleep disorder, arthralgia, fatigue, genitourinary tract infection, myalgia, blood pressure fluctuation, chronic cough, and excessive thirst.

The main limitation of our study includes the sample size required for a more accurate analytical approach. Despite the high prevalence of COVID-19 in Brazil and worldwide, HTLV-1 infection is still neglected and, thus, underdiagnosed, representing a challenge for most research groups [34]. This situation forces us to use this method, even imprecisely, as the only instrument available to measure this condition. It is also important to highlight that our findings were obtained from individuals infected with HTLV-1, extensively evaluated at the same outpatient clinic in the university hospital. This population reflects the worldwide prevalence in terms of sex, age, and associated manifestations. Another possible limitation refers to the fact that this study is retrospective. However, all questionnaires were completely filled out. Therefore, we believe that this study, supported by demographic and clinical data, is quite representative of the population with HTLV-1 infection.

In conclusion, we provide evidence that HTLV-1 status does not appear to worsen the outcome of COVID-19. The favorable evolution in most of our patients raises the question of whether HTLV-1 could have a protective effect against SARS-CoV-2 infection. Anxiety was the main risk factor for COVID-19 in HTLV-1 infection. It is essential that measures involving not only physical, but also mental, care should be implemented in public health programs for HTLV-1-infected individuals worldwide.

## Figures and Tables

**Table 1 pathogens-12-00242-t001:** Risk factors/comorbidity associated with COVID-19 in 51 Brazilian HTLV-1-infected individuals from Rio de Janeiro.

Demographic and Clinical Findings (n = 51)	COVID-19+ (n = 19)	Non-COVID-19 (n = 32)	*p* Value
Age, mean (SD), y	51.7 ± 13.0	56.5 ± 14.3	>0.05
Sex			
Female	12 (63.2%)	24 (75%)	>0.05
Male	7 (36.8%)	8 (25%)	
Race			
African descendant	10 (52.6%)	19 (59.4%)	>0.05
Caucasian	8 (42.1%)	13 (40.6%)	
Asian	1 (5.3%)	0 (0%)	
Risk factor			
Age ≥ 60	6 (31.6%)	15 (45%)	>0.05
Chronic neurological disease (HAM)	10 (52.6%)	22 (68.8%)	>0.05
Chronic obstructive lung disease	3 (15.8%)	1 (3.1%)	>0.05
Cardiovascular disease/hypertension	4 (21.1%)	6 (18.8%)	>0.05
Diabetes	2 (10.5%)	4 (12.5%)	>0.05
Anxiety	11 (57.9%)	5 (15.6%)	<0.05
Depression	4 (21.1%)	2 (6.3%)	>0.05
Sedentary lifestyle	12 (63.2%)	21 (65.6%)	>0.05
Obesity	3 (15.8%)	6 (18.8%)	>0.05
Current smoker	4 (21.1%)	5 (15.6%)	>0.05
COVID-19 vaccination	3 (15.8%)	4 (12.5%)	>0.05

**Table 2 pathogens-12-00242-t002:** Clinical characteristics of 18 co-infected Brazilian HTLV-1 patients with symptomatic COVID-19.

Clinical Characteristics of HTLV-1 Individuals	COVID-19+ (%)
COVID-19 symptoms	(n = 18)
Respiratory symptoms	15 (83.3%)
Ageusia	15 (83.3%)
Myalgia	15 (83.3%)
Fatigue	15 (83.3%)
Anosmia	13 (72.2%)
Headache	12 (66.7%)
Fever	11 (61.1%)
Cough	9 (50%)
Chills	9 (50%)
Somnolence	8 (44.4%)
Sore throat	7 (38.9%)
Inappetence	6 (33.3%)
Dyspnea	5 (27.8%)
Abdominal pain	4 (22.2%)
Coryza	4 (22.2%)
Mental confusion	3 (16.7%)
Desaturation (<95%)	3 (16.7%)
Low back pain	3 (16.7%)
Chest pain	3 (16.7%)
Pain in the lower limbs	2 (11.1%)
Burning eye sensation	1 (5.6%)
Treatment	
No treatment	2 (10.5%)
Symptomatic treatment at home	17 (89.5%)
Hospitalization	0 (0%)

**Table 3 pathogens-12-00242-t003:** Post-COVID-19 Conditions in 11 Patients.

Post-COVID-19 Conditions	(n = 11)
Ageusia and anosmia	6 (31.6%)
Worsening of anxiety	3 (15.8%)
Pain, burning, or subjective sensation of heaviness in lower limbs	3 (15.8%)
Dyspnea	5 (5.3%)
Backache	5 (5.3%)
Headache	5 (5.3%)
Sleep disorder	5 (5.3%)
Arthralgia	5 (5.3%)
Fatigue	5 (5.3%)
Genitourinary tract infection	5 (5.3%)
Pain in the lower limbs	5 (5.3%)
Myalgia	5 (5.3%)
Blood pressure fluctuation	5 (5.3%)
Excessive thirst	5 (5.3%)
Chronic cough	5 (5.3%)

## Data Availability

All data used in this study are anonymized and will be shared on request from any qualified investigator.

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
