# Peer review of "COVID-19 among People Living with HTLV-1 Infection in Rio de Janeiro, Brazil"

_pathogens, 2023, doi:10.3390/pathogens12020242_

Round 1

Reviewer 1 Report

The authors studied the impact of covid 19 on people infected with HTLV-1 and concluded that HTLV-1 did not increase the risk of lethal covid19 with solid data. This manuscript can be accepted if the following comments are properly addressed:

1. The language need to be improved. for example, line142, " All reported the use prevention measures", line172, " presented severe covid-19 due the presence of dyspnea and oxygen saturation below 95%", line192, "Although the high frequency of comorbidities/risk factors.... of the cases" and etc. 

2. we know that there have been different covid strains since 2020, what are the strains related to this study in Brazil? How would different strains affect your result?

Author Response

The authors studied the impact of covid 19 on people infected with HTLV-1 and concluded that HTLV-1 did not increase the risk of lethal covid19 with solid data. This manuscript can be accepted if the following comments are properly addressed:

  1. The language need to be improved. for example, line142, " All reported the use prevention measures", line172, " presented severe covid-19 due the presence of dyspnea and oxygen saturation below 95%", line192, "Although the high frequency of comorbidities/risk factors.... of the cases" and etc. 

Answer: The english language of the text was edited by MDPI.

  1. we know that there have been different covid strains since 2020, what are the strains related to this study in Brazil?

Answer: This information was included in the text -page 1 (lines 42-45) and page 2 (lines 46-51).

- How would different strains affect your result?

Answer: In general, our patients had a favorable evolution throughout the study period (2020-2022), regardless of the predominant viral strain. Patients who evolved to post-covid-19 conditions were not vaccinated, with most cases occurring in 2020, during the pandemic emergency. In addition, the group of patients studied has the unique characteristic of co-infection with COVID-19 and HTLV-1, making it difficult to associate with the predominant viral strain.

Reviewer 2 Report

There are no remarks to be made.

Author Response

English language and style are fine/minor spell check were edited by MDPI.

Reviewer 3 Report

Comments and Suggestions for Authors:

This manuscript by Puccioni-Sohler et al is a unique and timely paper containing a novel set of observations among a significant cohort of HTLV-1 infected individuals in a geographicl location which has had a large cohort of SARS-CoV-2 infection. This work thus and a unique opportunity to understand the interplay between these two viral infections in the context of the HTLV-1-altered immune system. The data are valuable and worthy of publication, although, as properly indicated by the authors, somewhat limited based on the numbers of co-infected individuals. This work is clearly important to publish.

The major focus of the paper is a comparison of symptoms in HTLV-1-infected patients with or without COVID. My major suggestion would be the addition to the discussion of some level of comparison with other reported series of COVID-19 infection, for example in Brazil, to try to identify  (or even suggest areas for further study) whether HTLV-1 co-infection is associated with any unique features of COVID-19. For example, is the persistent leg burning/pain a characteristic unique to the HTLV-1 infected COVID-19 patients and not to HTLV-1-negative COVID-19 patients?  Again, my suggestion strictly for a comparison to existing literature. At the very least, the issue of whether HTLV-1 co-infection leads to any changes in COVID-19 presentation or symptoms is worth a discussion paragraph of its own. For example, the complete absence of hospitalization is impressive, although the patient numbers are too small to suggest that there is truly any protective effect. On the other hand, the high rate of post-COVID symptoms suggests a need for careful long-term monitoring to determine if HTLV-1/SARS-CoV-2 co-infected individuals are indeed at higher risk for Long COVID. A comparison with results on long COVID in Brazil of the paper of Ferreira de Oliveria (PMID: 35908724)  and those of de Miranda et al (PMID: 35514142) and possibly others would be interesting and useful in beginning to address whether HTLV-1 may affect COVID, again understanding that the numbers are small..

Suggested  minor improvements to increase clarity of the manuscript:

1.     Please clarify lines 147-148:  There was no difference between HAM patients with or without corticosteroids 147 (p>0.05)”.  What is the parameter for which there was no difference between HAM patients with or without corticosteroids? Based on the order of the paper and of the upcoming discussion of Table 1, I presume that this means that there was no statistical difference in numbers of  those 6 individuals on steroids who did or did not get COVID, but important to clarify the rest of this point.

2.     Line 155.  The Table indicates a p value of <0.05 but the text says <0.01. Please clarify.

3.     Minor formatting point. The gender comparison on lines 160-162 is formatted as part of the Table, but I think more appropriately should be part of the text of section 3.2 as it is a description of data relevant to the Table, but not shown. (You may want to indication, “data not shown” for the gender breakdown.

Author Response

This manuscript by Puccioni-Sohler et al is a unique and timely paper containing a novel set of observations among a significant cohort of HTLV-1 infected individuals in a geographicl location which has had a large cohort of SARS-CoV-2 infection. This work thus and a unique opportunity to understand the interplay between these two viral infections in the context of the HTLV-1-altered immune system. The data are valuable and worthy of publication, although, as properly indicated by the authors, somewhat limited based on the numbers of co-infected individuals. This work is clearly important to publish.

The major focus of the paper is a comparison of symptoms in HTLV-1-infected patients with or without COVID. My major suggestion would be the addition to the discussion of some level of comparison with other reported series of COVID-19 infection, for example in Brazil, to try to identify  (or even suggest areas for further study) whether HTLV-1 co-infection is associated with any unique features of COVID-19. For example, is the persistent leg burning/pain a characteristic unique to the HTLV-1 infected COVID-19 patients and not to HTLV-1-negative COVID-19 patients?  

Answer: We included a comparison with other reported series of COVID-19 infection, for example in Brazil page 7 (lines 244-249),.

The persistent leg burning/pain, not compatible with leg syndrome ocurred in a patient with HAM (included in the page 5 - lines 199-200). It may be unique characteristic to the HTLV-1 infected COVID-19 patients but it is difficult to confirm considering two out three of these patients had already neurological involvement of HTLV-1 infection (HAM). Extremity paresthesias in the lower limbs are common complaints infection.Bu this discussion was included in the text (page 8 lines 320-321)

Again, my suggestion strictly for a comparison to existing literature. At the very least, the issue of whether HTLV-1 co-infection leads to any changes in COVID-19 presentation or symptoms is worth a discussion paragraph of its own. For example, the complete absence of hospitalization is impressive, although the patient numbers are too small to suggest that there is truly any protective effect. On the other hand, the high rate of post-COVID symptoms suggests a need for careful long-term monitoring to determine if HTLV-1/SARS-CoV-2 co-infected individuals are indeed at higher risk for Long COVID. A comparison with results on long COVID in Brazil of the paper of Ferreira de Oliveria (PMID: 35908724)  and those of de Miranda et al (PMID: 35514142) and possibly others would be interesting and useful in beginning to address whether HTLV-1 may affect COVID, again understanding that the numbers are small..

Answer: We also included the comparison with the studiessuggested by the reviewer ( page 8  lines 302-310)  

Suggested  minor improvements to increase clarity of the manuscript:

  1. Please clarify lines 147-148:  “There was no difference between HAM patients with or without corticosteroids 147 (p>0.05)”.  What is the parameter for which there was no difference between HAM patients with or without corticosteroids? Based on the order of the paper and of the upcoming discussion of Table 1, I presume that this means that there was no statistical difference in numbers of  those 6 individuals on steroids who did or did not get COVID, but important to clarify the rest of this point.

Answer: We clarified the text that” There was no statistical difference in the frequency of COVID-19 between HAM patients treated with corticosteroids (n=6) in comparison to the non-treated (n=4) group (p>0.05).” in the page 4 (lines 170-172)

  1. Line 155.  The Table indicates a p value of <0.05 but the text says <0.01. Please clarify.

Answer: It was corected in the text p value of <0.05 (page 4, line 170)

  1. Minor formatting point. The gender comparison on lines 160-162 is formatted as part of the Table, but I think more appropriately should be part of the text of section 3.2 as it is a description of data relevant to the Table, but not shown. (You may want to indication, “data not shown” for the gender breakdown.

 Answer: The text about gender comparison was included in the the text of section 3.2 and statistical significance (page 4 lines 163-166)
